# Fabrication and Sensing Application of Phase Shifted Bragg Grating Sensors

**DOI:** 10.3390/ma15103720

**Published:** 2022-05-23

**Authors:** Xiaoyan Sun, Li Zeng, Youwang Hu, Ji’an Duan

**Affiliations:** State Key Laboratory of High Performance Complex Manufacturing, College of Mechanical and Electrical Engineering, Central South University, Changsha 410083, China; sunxy@csu.edu.cn (X.S.); 183701037@csu.edu.cn (L.Z.); duanjian@csu.edu.cn (J.D.)

**Keywords:** fiber Bragg grating, phase-shifted fiber Bragg grating, fiber optics sensors, femtosecond laser

## Abstract

As a special kind of Bragg grating, phase-shifted fiber Bragg grating (PS-FBG) has attracted extensive attention because of its extremely narrow transmission window and excellent sensing performance. The main purpose of this manuscript is to discuss the PS-FBG with special sensing characteristics and explore the influence of different inscription technologies on the sensing characteristics of PS-FBG by comparing the existing inscription methods. The sensing characteristics, advantages and disadvantages of PS-FBG with different structures are analyzed.

## 1. Introduction

Optical fiber sensors are an important part of modern sensors. They have the advantages of small volume, anti-electromagnetic interference, multi-parameter sensing, high safety and remote monitoring [1,2,3,4,5,6,7,8,9,10]. These characteristics make them very suitable for application in the fields of biomedicine, the petrochemical industry, aerospace, etc. [11,12,13,14]. Fiber Bragg grating (FBG) is an important part of optical fiber sensors which has many advantages such as compact structure, distributed measurement, small transmission loss and anti-electromagnetic interference [15]. In addition, conventional FBG is not sensitive to bending and external refractive index (RI). This makes the measurement of strain, temperature and other parameters more convenient and accurate.

With the progress of FBG inscribe technology, many innovative forms of Bragg gratings have been realized, such as chirped FBG [16,17,18,19,20], titled FBG [21,22,23,24] and phase-shifted fiber Bragg grating (PS-FBG) [25,26,27,28]. Among them, PS-FBG has attracted extensive attention [29]. Different from conventional gratings, there are one or more narrow transmission windows in the spectrum of PS-FBG [30,31,32]. PS-FBGs not only have the advantages of FBGs but also have many special sensing characteristics which allow different phase shift introduction methods. In addition, the transmission windows have higher sensing accuracy. Therefore, they have high application value in various high-performance sensors [33,34,35].

At present, researchers have developed a variety of technologies for the production of PS-FBG, including localized micro-strain and heat [36,37,38,39], mobile fiber scanning beam method [40], arc discharge erasing technique [41], shielded phase mask method [42] and electro-optic amplitude modulated laser pulses [43,44], etc. The sensing characteristics of PS-FBG inscribed by different technologies are different. PS-FBG inscribed by some of these methods has excellent sensing performance due to its special phase-shift introduction mode. Although the sensing object and sensitivity of PS-FBG inscribed by other methods are the same as that of ordinary FBG, the achievable resolution for measurement is considerably improved, due to the narrow bandwidth.

In this paper, we review the recent advances in the sensing applications of PS-FBGs. Three kinds of PS-FBG with high-performance sensing characteristics inscribed by different processing methods are introduced. In Section 2, The effects of different parameters on PS-FBG spectra were analyzed by the transfer matrix method. In Section 3, we introduce the PS-FBG sensors with unique sensing characteristics fabricated by different processing methods.

## 2. Analysis of Spectral Characteristics

The transmission matrix method is used to study the influence of different grating parameters on the PS-FBG spectrum [45]. There are forward mode and backward mode in the FBG. A section of grating can be regarded as a dual-port device. Set *z_i_* as the input end, *z_i_*_+1_ as the output end, *A*(*z_i_*) and *B*(*z_i_*) as the amplitudes of the input, and *A*(*z_i_*_+1_) and *B*(*z_i_*_+1_) as the amplitudes of output. The solution of the coupled-mode equation in matrix form is
(1)$$\left[\begin{array}{c} A(z_{i+1})\\ B(z_{i+1}) \end{array}\right]=Fz_{i}z_{i+1}\left[\begin{array}{c} A(z_{i})\\ B(z_{i}) \end{array}\right]$$
(2)$$Fz_{i}z_{i+1}=\left[\begin{array}{cc} s_{11} & s_{12}\\ s_{21} & s_{22} \end{array}\right]$$
where $Fz_{i}z_{i+1}$ is the transfer matrix, and the matrix elements of the S matrix are obtained by using standard coupling modes. The derivation process has been detailed in previous reports [45]. For standard FBG, *B*(*z_i_*_+1_) = 0 [46]
(3)$$B(z_{i})=-\frac{s_{21}}{s_{22}}A(z_{i})$$
(4)$$A(z_{i+1})=(s_{11}-\frac{s_{12}s_{21}}{s_{22}})A(z_{i})$$

Then, the reflectance and transmittance of FBG are, respectively,
(5)$$R={\left|\frac{B(z_{i})}{A(z_{i})}\right|}^{2}={\left|\frac{s_{21}}{s_{22}}\right|}^{2}$$
(6)$$T={\left|\frac{A(z_{i+1})}{A(z_{i})}\right|}^{2}={\left|s_{11}-\frac{s_{12}s_{21}}{s_{22}}\right|}^{2}$$

Assuming that there are *M* − 1 phase shift points in PS-FBG, it can be regarded as being connected by *M* segments of gratings with the same period but with phase shift, thus, the total input and output relationship is
(7)$$\left[\begin{array}{c} A(z_{M})\\ B(z_{M}) \end{array}\right]=F_{M}F_{M-1}\cdot \cdot \cdot F_{i}\cdot \cdot \cdot F_{1}\left[\begin{array}{c} A(z_{1})\\ B(z_{1}) \end{array}\right]=F\left[\begin{array}{c} A(z_{1})\\ B(z_{1}) \end{array}\right]$$

According to Formulas (5)–(7), the spectrum of PS-FBG can be calculated.

The different refractive index modulation, the magnitude of phase shift and the position of phase shift point are compared respectively. The reflection spectra of FBGs with different refractive index modulation and the same other parameters are shown in Figure 1a. The full width at half maximum (FWHM) will increase with the increase of refractive index modulation. π phase-shifted FBGs with different refractive index modulation are shown in Figure 1b, and the introduction of phase shift opens a narrow transmission window in the FBG resonance peak. With the increase of refractive index modulation, the FWHM of the transmission window decreases.

The different values of phase shifts are simulated, the spectra are shown in Figure 2. The period, modulation and phase shift points of the grating are the same. The position of the transmission window is determined by the phase shift value. When the phase shift is π, it is located at the center.

In addition, different positions of phase shift points will also affect the phase shift peaks. If the length of the grating is L, introduce π phase shift at 0.25 L, 0.375 L and 0.5 L respectively. The calculated spectrum is shown in Figure 3. The results show that the position of the phase shift point will not affect the position of the phase shift peak, but the closer the phase shift point is to the midpoint of the gate region, the higher the visibility of the phase shift peak is. Therefore, when processing PS-FBG sensors, the phase shift point should be at the midpoint of the grating area.

## 3. Phase Shifted Fiber Bragg Grating Sensors

### 3.1. Phase Shift Introduced by Special Structure or Materials

In recent years, researchers have proposed a variety of preparation methods for PS-FBG and achieved excellent sensing performance. Introducing a special structure or filling special materials in the middle of the FBG grating region can not only introduce phase shift, but can also give special sensing characteristics. For example, the fusion region is introduced to realize the two parameters measurement of temperature and strain [47], the taper structure is introduced to realize high-sensitivity strain measurement [48], the magnetic fluid is infiltrated to realize the magnetic field sensing [49], and the photosensitive glue is filled to realize the three parameters measurement [50]. When PS-FBG is fabricated in this way, the same technology as conventional FBG can be used when inscribing gratings, such as the phase mask method and the femtosecond laser direct writing method. Compared with shielding phase mask [42] and electro-optic amplitude modulation [44], the methods mentioned above reduced the manufacturing difficulty of gratings.

Using fusion splicing technology and femtosecond laser phase mask technology can fabricate PS-FBG with dual parameter sensing [47,51]. The device is shown in Figure 4a. The fabrication processing is simple and efficient: cut one section of SMF into two sections and then splice them together again. Finally, an FBG is inscribed in the fiber core by the phase mask technique. In the process of optical fiber fusion, due to dopant diffusion, glass structure change and residual stress relaxation, the effective refractive index of the fusion region with a length of hundreds of microns will be reduced, thus the fusion region acts as a phase shifter [52].

Due to the non-uniformity of the fusion structure, the phase shift changes differently with the increase in temperature and strain. By measuring the central wavelength and loss difference of the two dips, the simultaneous measurement of temperature and strain can be realized. The test range of temperature and strain are −50–150 °C and 0–2070 με, respectively. The sensitivity of temperature and strain is 10.3 pm/°C and −1.23 pm/με, respectively. From Equation (1), the temperature and strain can be differentiated simultaneously by measuring the changes in central wavelength and the loss difference between the two dips when the PS-FBG is placed in an environment with simultaneous changes in temperature and strain. As a function of the variation of temperature (Δ*T*) and strain (Δ*ε*), the center wavelength shift of the two dips (Δ*λ*) and loss difference (Δ*L*) can be written as:
(8)$$\left[\begin{array}{c} \Delta \lambda \\ \Delta L \end{array}\right]=\left[\begin{array}{cc} K_{\lambda \varepsilon } & K_{\lambda }{}_{T}\\ K_{L\varepsilon } & K_{LT} \end{array}\right]\left[\begin{array}{c} \Delta \varepsilon \\ \Delta T \end{array}\right]$$
where *K_λT_* and *K_LT_* are the temperature sensitivities, *K_λε_* and *K_Lε_* are the strain sensitivities, respectively. A sensitivity matrix for simultaneous measurement of temperature and strain can be derived as
(9)$$\left[\begin{array}{c} \Delta \varepsilon \\ \Delta T \end{array}\right]=\frac{1}{M}\left[\begin{array}{cc} K_{LT} & -K_{\lambda }{}_{T}\\ -K_{L\varepsilon } & K_{\lambda \varepsilon } \end{array}\right]\left[\begin{array}{c} \Delta \lambda \\ \Delta L \end{array}\right]$$
where *M* = *K_LT_K_λε_*
*− K_Lε_K_λT_* is the determinant of the coefficient matrix. The values of *K_LT_*, *K_λε_, K_Lε_* and *K_λT_* can be obtained through the sensitivity test to temperature and strain. Thus, the matrix can be written as:(10)$$\left[\begin{array}{c} \Delta \varepsilon \\ \Delta T \end{array}\right]=\frac{1}{5.74\times 10^{-3}}\left[\begin{array}{cc} 2.28\times 10^{-3} & 10.3\\ 2.28\times 10^{-3} & -1.23 \end{array}\right]\left[\begin{array}{c} \Delta \lambda \\ \Delta L \end{array}\right]$$

PS-FBG with high strain sensitivity can be fabricated by preparing the taper region by arc discharge and writing FBG on both sides of the taper region [48]. Firstly, the commercial SMF with the coating removed is installed on the fusion splicer, and a micron taper structure is prepared on the optical fiber by using the discharge function of the machine. Then, two FBGs with the same period are inscribed on both sides of the taper structure by femtosecond laser line-by-line technology.

The device is shown in Figure 5. The phase shift is caused by the refractive index change of the taper when the axial strain is applied. The phase shift can be changed by varying the strain and that strain sensitivity could be enhanced by using a thinner taper. A device without taper and two devices with minimum core diameters of 4.6 μm and 2.2 μm were fabricated. During the test, with the increase of strain, the phase shift peaks and FBG resonance peaks have a significant redshift. Compared with the device without a taper, the redshift of phase shift peaks with the taper is greater than that of the FBG resonance peaks. The sensitivity of the three samples is 0.949 pm/με, 2.37 pm/με and 4.59 pm/με, respectively.

PS-FBG sensitive to magnetic field intensity can be fabricated by infiltrating magnetic fluid into the middle of the FBG [49]. A high reflection FBG with a length of 12 mm was inscribed using a dual frequency argon laser. The 3 dB bandwidth is 0.3 nm and the reflectivity is 25 dB. Then, fiber is cleaved in the middle of FBG. The two FBG slices are encapsulated with ceramic ferrules and the ends of the FBG sections with a length of 0.2 mm outside the ceramic ferrule are exposed for convenience of observation. Two ceramic ferrules containing FBG sections are connected by a sleeve with a longitudinal slit of -0.5 mm, which can be used to inject magnetic fluid. Then, one ceramic ferrule is fixed on the translation stage, and the other remains fixed. By pushing and pulling the ceramic insert on the translation table, a micron controllable air gap can be introduced at both ends of the optical fiber. Finally, magnetic fluid is injected into the air gap, and a PS-FBG with narrow transmission peak and extinction ratio of 7 dB is formed.

The device is shown in Figure 6. The magnetic fluid is EMG 605 water-based ferromagnetic fluid. The volume concentration of Fe_3_O_4_ nanoparticles is 3.6%. In an environment without a magnetic field, Fe_3_O_4_ nanoparticles are evenly distributed in the solution due to Brownian motion. When the magnetic field intensity increases, the nanoparticles form chain clusters along the direction of the magnetic field, thus changing the RI of the magnetic field [53,54,55]. The air gap introduces a phase shift in the FBG, and the magnetic fluid is injected into the gap. The change of refractive index of magnetic fluid will lead to the change of phase shift. The experimental device is shown in Figure 7. The magnetic field is perpendicular to the axial direction of the optical fiber, and the magnetic field intensity increases from 0 Oe to 120 Oe. With the increase of magnetic field intensity, the central wavelength of phase shift peak shifts to the long wavelength direction at a linear rate of 2.42 pm/Oe, and the intensity sensitivity is 0.044 dB/Oe. The device based on PS-FBG is shorter and has higher resolution than other types of MF-coated fiber sensors [56,57,58,59]. It is worth noting that magnetic fluid has magnetic saturation effect, which will limit the measurement range. In addition, magnetic-fluid based devices usually need temperature compensation [55].

Yang presented a structure similar to the magnetic fluid-infiltrated PS-FBG. The difference is that magnetic fluid is replaced by a photosensitive adhesive. Due to the curing characteristics of photosensitive adhesive, the manufacturing process of the device is simpler and the volume is smaller [50]. Firstly, an FBG is inscribed in the standard SMF with a central wavelength of 1550 nm and a grating length of 15 mm by 193 nm excimer laser. Then, the fiber is cleaved in the middle of the FBG and the two sections are clamped to the fusion splicer. The alignment of optical fiber and manufacturing gap between two ends can be easily realized by fusion splicer. Finally, photosensitive glue is dropped and cured. The device is shown in Figure 8. The distance between the two halved FBGs is determined to be approximately 57 μm. Because the periodic refractive index distribution of FBG is interrupted, a phase shift is formed.

The closed Fabry–Pérot interferometer can measure the changes in temperature, pressure and RI caused by the change of cavity length, while the wavelength of the phase shift peak of PS-FBG is only sensitive to temperature and the intensity of the structural phase shift spectrum is only insensitive to pressure. Due to the difference in sensing characteristics between EFPI and PS-FBG, the device can realize the differential measurement of temperature, pressure and RI by using the methods of wavelength modulation and intensity modulation. The sensitivity obtained from the experiment is shown in Equation (11). This integrated sensor has high research value in the field of multi parameter measurement.
(11)$$\left[\begin{array}{c} \Delta n\\ \Delta T\\ \Delta P \end{array}\right]=-\frac{1}{1.61}\left[\begin{array}{ccc} 0 & -0.19 & -0.01\\ 0 & -259.05 & 0\\ -1.98 & 179.99 & 2.20 \end{array}\right]\left[\begin{array}{c} \Delta \lambda_{EFPI}\\ \Delta \lambda_{PS}\\ \Delta K_{PS} \end{array}\right]$$

Femtosecond laser micromachining technology can process microchannels in the optical fiber to make the liquid or gas enter the fiber core, which makes PS-FBG sensitive to the change in the surrounding refractive index. In addition, the phase shift value of PS-FBG fabricated by this method is not affected by temperature, thus, the interference of temperature can be easily eliminated.

### 3.2. PS-FBG Sensors Fabricated by Femtosecond Laser Micromachining Technology

Our research group presented a method of PS-FBG fabrication by femtosecond laser selective etching, which can accurately control the phase shift [60]. Firstly, an FBG was inscribed in SMF using the femtosecond laser direct writing technique. Then, a track was carved in the middle of the Bragg grating with a femtosecond laser. Finally, by immersing the optical fiber in a 5% hydrofluoric acid (HF) solution, the area modified by a femtosecond laser is easier to be etched, so a microchannel can be etched along the path crossed by the laser. Using a broadband light source and a spectrometer to monitor the spectral changes in the etching process in real-time can accurately control the phase shift.

The device is shown in Figure 9. A micro-channel passes through the fiber core, which introduced a phase shift in the fiber grating. More importantly, it exposes the fiber core to the external environment, which makes the phase shift affected by the external refractive index. The refractive index sensitivity of the device is 2.526 nm/RIU and −111 dB/RIU, in the range of 1.364–1.413. The temperature change has little effect on the phase shift.

Combining femtosecond laser micromachining technology with fiber fusion technology, PS-FBG with high refractive index sensitivity can be fabricated by making bubbles in the grating on FBG [61]. Firstly, an excimer laser is used to process an FBG with a central wavelength of 1578 nm using the phase mask method, and the FBG is cut from the midpoint by a fiber cutter. Then, a micropore is processed in the center of a cutting section by a femtosecond laser, and the two sections are spliced. When the air in the micropore is suddenly heated, it will expand into a bubble. Finally, a micro-channel through the center of the bubble was fabricated by fs laser drilling technology. Micro-channels can make the liquid enter the bubble and cause a change of phase shift.

The schematic diagram of the device is shown in Figure 10. The length of FBG is 5 mm. The diameters of bubble and micro-channel are about 50 μm and 14 μm, respectively. A phase shift peak around the center of FBG stop band at 1578.8 nm with FWHM of 30 pm. In the RI range of 1.400–1.420, the refractive index sensitivity is 9.9 nm/RIU. The temperature sensitivity of the phase shift peak is 10.19 pm/°C, and the temperature has little effect on the phase shift. This device has great sensing potential. Different sensing functions can be realized by injecting liquids with different characteristics into a bubble.

The insertion of hollow fiber (HCF) into the FBG gate area can also achieve good results [62]. Firstly, a hollow fiber with an inner diameter of 75 um is spliced between two pieces of SMF, then a pair of FBGs are inscribed in the SMF on both sides of the HCF by femtosecond laser line-by-line method. The two FBGs have the same Bragg resonant wavelength of 1550 nm. Finally, a pair of microchannels with a diameter of 15 um is drilled on the hollow fiber by femtosecond laser.

The device is shown in Figure 11. The hollow fiber introduces a special phase shift in the FBG, and the microchannel can make the external air enter the hollow cavity. The change of air pressure will change the refractive index of air, resulting in the change of phase shift. Through the experimental test, in the range of 0–2.25 MPa, the change of phase shift wavelength is linearly related to the change of gas pressure. When the hollow cavity length is 88.3 μm, the gas pressure sensitivity achieved 1.22 nm/Mpa. In addition, the temperature sensitivity of the phase shift peak is 8.92 pm/°C.

### 3.3. PS-FBG Sensors with High Birefringence

FBG inscribed with a high-intensity fs laser has high birefringence [43]. The phase mask technology and line-by-line inscription by an fs laser have a long strip in the refractive index modulation region, which can bring a higher birefringence effect. Using this feature, torsion and polarization sensing can be realized.

Huang first proposed using fs laser line-by-line technology to inscribe PS-FBG, and realized torsion sensing by using the high birefringence brought by line-by-line technology [63]. The processing path of PS-FBG processed by line-by-line technology is shown in Figure 12a. PS-FBG consists of two parts with a spacing of ΔΛ and FBG with the same period. FBG is composed of linear waveguides penetrating the fiber core. The grating period is determined by the spacing of adjacent linear waveguides, and the magnitude of phase shift is determined by the gap of two FBGs. Therefore, compared with the fs laser point-by-point technology, this technology has lower requirements for the accuracy of optical fiber calibration and positioning, and the value of ΔΛ can be controlled more accurately. More importantly, the RI modulation region of the linear waveguide is distributed along an axial direction of the cross-section of the optical fiber, thus it cannot be guaranteed that the RI modulation distribution region is symmetrical about the central point of the optical fiber core. The corresponding stress distribution in the optical fiber is also asymmetric. Therefore, it is easy to introduce birefringence into the optical fiber.

The length of the linear waveguide is 8 μm. The grating length is 1.5008 mm. The larger birefringence introduced into the fiber after inscribing the grating, the intensity change of the transmission peak can reach 2.265 dB under different polarization states, which is enough for sensing demodulation. When the polarization state of the light source is ~45° the spectrum diagram is shown in Figure 12b, and the resonance peak is split into two peaks with equal transmittance. The sensitivity can be amplified by using the difference in the intensity of two transmission peaks to realize torsional sensing.

The torsion test device is shown in Figure 13. One side of the optical fiber is fixed, and the device is twisted clockwise and counterclockwise by rotating the other side. The corresponding torsion sensitivity of the device in the linear fitting region of 140° to 240° clockwise and 100° to 210° counterclockwise is 963.53 dB/(rad/mm) and 1032.71 dB/(rad/mm) respectively. In addition, temperature and strain have little effect on the transmission peak intensity. This device has high torsional sensitivity, small size (only ~1.72 mm), an extremely strong and simple structure, and is very attractive in practical applications.

PS-FBG with high birefringence can also be fabricated by the femtosecond laser phase mask method. Halstuch proposed an overlapped grating structure, in which two FBGs with similar central wavelengths are inscribed at the same position, which has a high-quality PS-FBG spectrum. The measured 3 dB bandwidth is only 40 μm. The structure is shown in Figure 14a. FBG is inscribed by fs phase mask technology. After the inscription of the first FBG, appropriate tension is applied to the optical fiber to produce slight strain and inscribed the second FBG. Due to the application of pre-strain, the second FBG shifts slightly. By controlling the applied tension, different phase-shifting structures can be characterized. Because the two FBGs are inscribed in the same position, the length of PS-FBG is reduced by half. The high sensitivity of the strain-assistant PS-FBG to polarization may result from the nonlinear index change and the high birefringence induced by the fs laser. The influence of polarized light on the PS-FBG spectrum is studied by using a polarization controller. There is an obvious polarization correlation in spectral. When the linear polarization is 45°, a spectrum similar to that in Figure 12b appears.

In addition to the strain-assistant method, our research group proposed a tilt-assistant method that could also achieve the same result [64]. The schematic diagram is shown in Figure 14b. After using the conventional fs laser phase mask technology to inscribe FBG 1, the rotating platform tilts the fiber and inscribes FBG 2. This method is very simple and accurate.

The pre-prepared fusion region or taper structure is relatively simple to process and can also improve the sensing performance. Infiltrating special materials in the middle of FBG can give the grating unique sensing characteristics, but this method will reduce the mechanical strength of the sensor and increase the volume of the device, and the scope of application will be limited. Femtosecond laser micromachining technology can expose the fiber core to the external environment to prepare refractive index-sensitive PS-FBG, but the processing is relatively difficult. In addition, the devices prepared in this way have high expansibility and are also suitable for material filling. The high birefringence brought by the femtosecond laser line-by-line method and phase mask method can realize the measurement of polarization and expand the direction of FBG sensing.

## 4. Summary

In this review, we have analyzed the spectral characteristics of the PS-FBG transmission window and introduced different types of PS-FBG sensors. Introducing special structures or filling special materials into the FBG midpoint can create specific sensing performance. The RI-sensitive phase shift peak can be obtained by cutting off the FBG and exposing the fiber core to the external environment. PS-FBG with high birefringence is prepared by the femtosecond laser line-by-line method or phase mask method to realize polarization sensing. For the sensor based on PS-FBG, the device with unique sensing performance can be prepared through the design of phase shift introduction mode. In addition, the extremely narrow transmission window and compact structure make the PS-FBG sensor of great research value.

## Figures and Tables

**Figure 1 materials-15-03720-f001:**
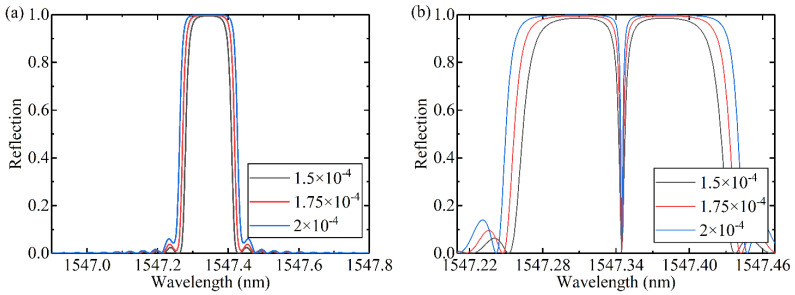
The reflection spectra of (**a**) FBGs and (**b**) π-PS-FBGs with different refractive index modulation.

**Figure 2 materials-15-03720-f002:**
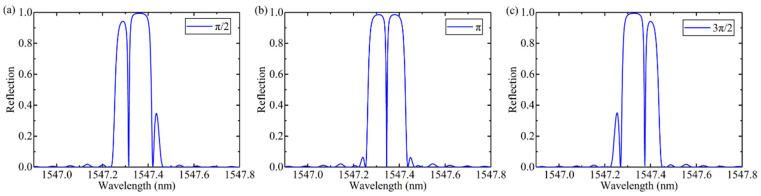
Reflection spectra of FBG (**a**) and phase-shifted FBG for three different values of phase shift (**a**) π/2, (**b**) π, and (**c**) 3π/2.

**Figure 3 materials-15-03720-f003:**
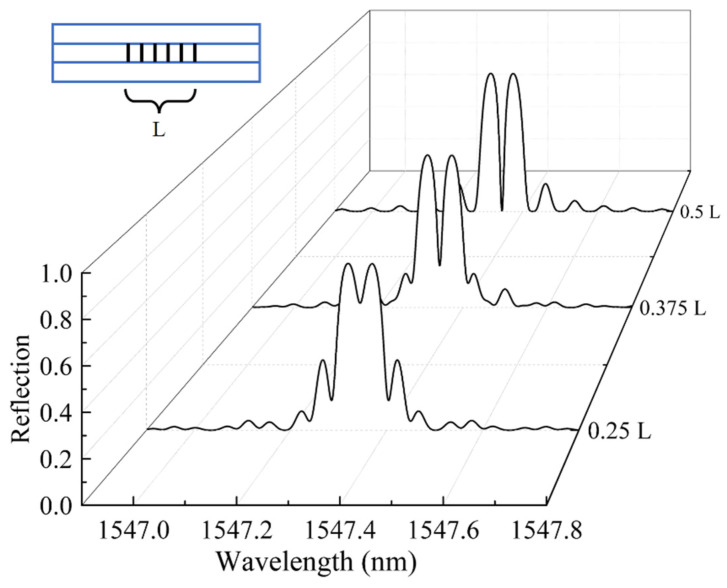
Reflection spectra of three π-PS-FBGs with different phase shift position.

**Figure 4 materials-15-03720-f004:**
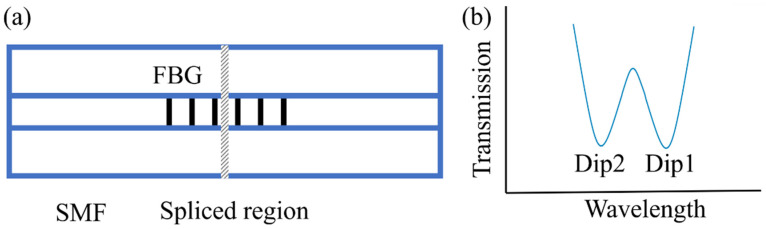
(**a**) Schematic diagram of the PS-FBG, modulated by a fusion spliced region. (**b**) The spectrum diagram of the PS-FBG fabricated by fusion splicing technology.

**Figure 5 materials-15-03720-f005:**
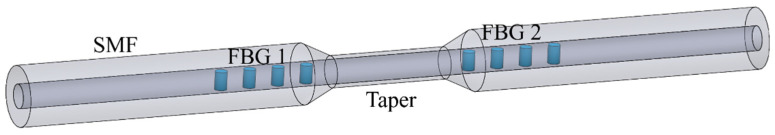
The schematic diagram of tapered PS-FBG.

**Figure 6 materials-15-03720-f006:**
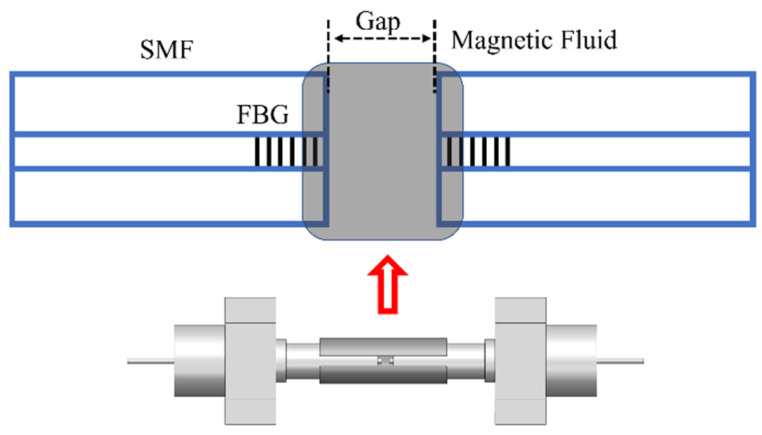
The schematic diagram of the magnetic fluid-infiltrated PS-FBG.

**Figure 7 materials-15-03720-f007:**
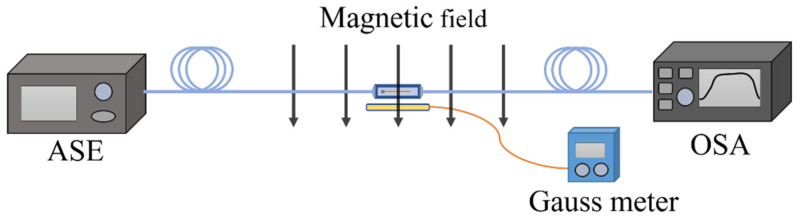
Magnetic field intensity testing device.

**Figure 8 materials-15-03720-f008:**
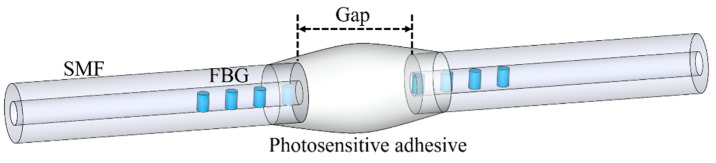
The schematic diagram of the photosensitive adhesive-infiltrated PS-FBG.

**Figure 9 materials-15-03720-f009:**
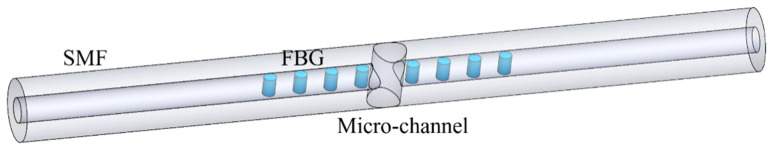
Schematic diagram of the designed PS-FBG, modulated by a micro-channel.

**Figure 10 materials-15-03720-f010:**
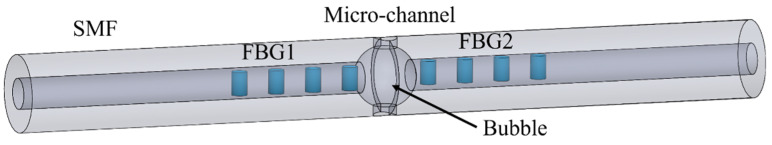
Schematic diagram of the designed PS-FBG in single-mode fiber.

**Figure 11 materials-15-03720-f011:**
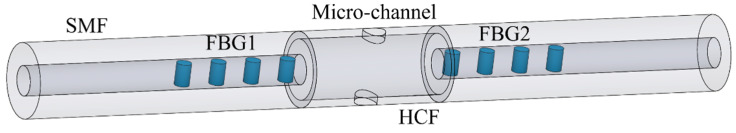
Schematic diagram of the PS-FBG, modulated by a hollow cavity.

**Figure 12 materials-15-03720-f012:**
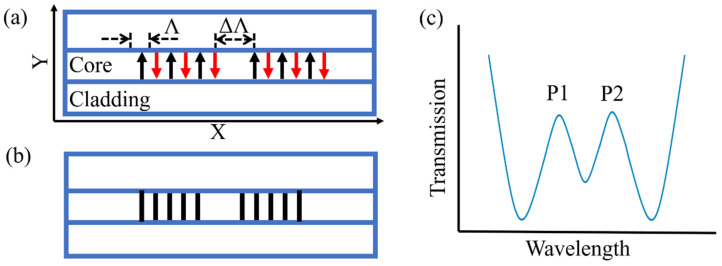
(**a**) The processing path of PS-FBG processed by line-by-line technology. (**b**) Schematic diagram of the designed PS-FBG. (**c**) The spectrum diagram when the original polarization state of input light is ~45°.

**Figure 13 materials-15-03720-f013:**
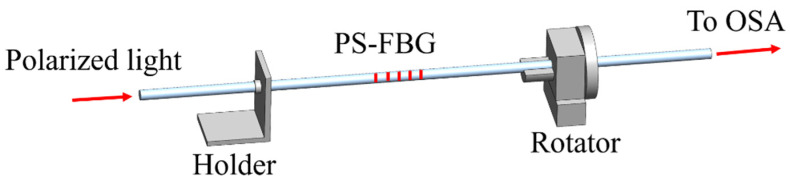
Torsion test device.

**Figure 14 materials-15-03720-f014:**
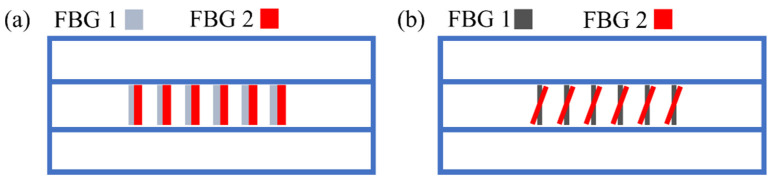
(**a**) The schematic diagram of the strain-assistant structure. (**b**) The schematic diagram of the tilt-assistant structure.

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
