# Peer review of "Fabrication and Sensing Application of Phase Shifted Bragg Grating Sensors"

_materials, 2022, doi:10.3390/ma15103720_

Round 1
Reviewer 1 Report
See the file attached.

Author Response
Thank you very much for your comments and suggestions. We are happy to have such excellent advice for improving the paper’s quality. With our best efforts, we have made changes by following the comments. This report summarizes our understanding of the recommendations and the changes we have made in response.

Reviewer 2 Report
In this paper, the authors present one review about phase-shifted fiber Bragg grating with special sensing characteristics and explore the influence of different inscription technologies on the sensing characteristics of PS-FBG by comparing the existing inscription methods. The sensing characteristics, advantages, and disadvantages of PS-FBG with different structures are analyzed. Phase shift FBG has some applications such review is nice for community interest in this topic. This review is clear, concise, and suitable for the scope of the journal. One main suggestion is supplied:
Suggest the authors add one application part, the title includes fabrication and application. Now the main focus is on the fabrication method.
Author Response

(The authors gave the same response as above.)

Reviewer 3 Report
Authors presented a review of phase shifted FBGs. In general, the review is interesting, well organized and can help researchers in the optical fiber sensing fields. However, some adjustments are needed prior to the publication.
- There are many sentences in which a reference should be included. Such as the one about the transfer matrix method.
- Authors should include the spectra of the PS-FBGs obtained from the different methods discussed throughout the paper.
- Authors should include a critical comparison between the different PS-FBG fabrication methods discussed
Author Response

(The authors gave the same response as above.)

Round 2
Reviewer 1 Report
In the reviewed manuscript, the authors addressed the comments reported after the first round of revision, apart from comment 11 in the cover letter where it is not clear the response.
The authors explained the meaning of the numbers in the matrix reported in Equation 1 of the manuscript only in the cover letter, but, in my opinion, they should add the explanation also in the manuscript.
Apart from the last comment above, I consider the manuscript suitable for publication in this journal.
Author Response
Thank you very much for your comments and suggestions. We explained comment 11 more clearly and added the explanation to the manuscript.
